# Genomic Characterization of *Staphylococcus aureus* in Wildlife

**DOI:** 10.3390/ani13061064

**Published:** 2023-03-15

**Authors:** Carmen Martínez-Seijas, Patricia Mascarós, Víctor Lizana, Alba Martí-Marco, Alberto Arnau-Bonachera, Eva Chillida-Martínez, Jesús Cardells, Laura Selva, David Viana, Juan M. Corpa

**Affiliations:** 1Biomedical Research Institute, PASAPTA-Pathology Group, Facultad de Veterinaria, Universidad Cardenal Herrera-CEU, CEU Universities, C/Tirant lo Blanc 7, Alfara del Patriarca, 46115 Valencia, Spain; 2Servicio de Análisis, Investigación, Gestión de Animales Silvestres (SAIGAS), Facultad de Veterinaria, Universidad Cardenal Herrera-CEU, CEU Universities, C/Tirant lo Blanc 7, Alfara del Patriarca, 46115 Valencia, Spain; 3Wildlife Ecology & Health Group (WE&H), Universitat Autònoma de Barcelona (UAB), Edifici V, Travessera dels Turons, Bellaterra, 08193 Barcelona, Spain

**Keywords:** *Staphylococcus aureus*, wild animal, whole genome sequencing, MLST, antibiotic resistance, virulence genes, MRSA, reservoir

## Abstract

**Simple Summary:**

*Staphylococcus aureus* is both a commensal and opportunistic pathogen that colonizes human and animal skin and mucosal membranes. It is widely acknowledged that livestock can act as a reservoir for human strains and genetic traits, but less is known about the epidemiological significance of wild animals. In the present work, the *S. aureus* isolated from wild animals, hunters, and hunting auxiliary animals of Eastern Spain are described and characterized. The genomic structure of the population of the sequenced isolates showed a significant degree of diversity, raising the question of these strains’ origin and evolution. In addition, genomic characterization revealed the presence of relevant genes in the isolates belonging to wild animals, including resistance genes, as well as superantigens implicated in the pathogenicity of certain human *S. aureus* infections. These findings highlight the role of wild animals as a reservoir for some clinically relevant *S. aureus* strains and the importance of wildlife surveillance because the environment-animals-humans interaction impacts the transmission and evolution dynamics of *S. aureus*.

**Abstract:**

*Staphylococcus aureus* is an opportunistic multi-host pathogen that threatens both human and animal health. Animals can act as a reservoir of *S. aureus* for humans, but very little is known about wild animals’ epidemiological role. Therefore, in this study, we performed a genomic characterization of *S. aureus* isolates from wildlife, hunters, and their auxiliary hunting animals of Eastern Spain. Of 20 different species, 242 wild animals were examined, of which 28.1% were *S. aureus* carriers. The common genet, the Iberian ibex, and the European hedgehog were the species with the highest *S. aureus* carriage. We identified 30 different sequence types (STs), including lineages associated with wild animals such as ST49 and ST581, multispecies lineages such as ST130, ST398, and ST425, and lineages commonly isolated from humans, including ST1 and ST5. The hunters and the single positive ferret shared ST5, ST398, or ST425 with wild animals. In wildlife isolates, the highest resistance levels were found for penicillin (32.8%). For virulence factors, 26.2% of them carried superantigens, while 14.8% harbored the immune evasion cluster (IEC), which indicates probable human origin. Our findings suggest that wild animals are a reservoir of clinically relevant genes and lineages that could have the potential to be transmitted to humans. These data support the notion that wildlife surveillance is necessary to better understand the epidemiology of *S. aureus* as a pathogen that circulates among humans, animals, and the environment.

## 1. Introduction

*Staphylococcus aureus* is both a commensal and opportunistic pathogen that colonizes human and animal skin and mucosal membranes [1]. Clinical presentations include a large panel of infections ranging from dermal diseases to deep infections and septicemia [2]. *S. aureus* has a clonal population structure in which certain clonal complexes are associated with a very specific host, while others have a wider range and are able to jump between species [1,3]. As it is a multi-host pathogen that threatens both human and animal health, it is essential to understand the genetic basis of transmissibility, adaptation, and host pathogenesis.

The success of *S. aureus* as a pathogen result from its exceptional genetic plasticity, which allows the acquisition of new resistance and virulence mechanisms [4]. Mobile genetic elements (MGEs), such as Staphylococcal Cassette Chromosome *mec* (SCC*mec)*, are one of the key components in *S. aureus*’ adaptability. SCC*mec* integration leads to methicillin-resistant *Staphylococcus aureus* strains (MRSA) emerging. Since its discovery, MRSA has been paid attention to because of its clinical repercussions as one of the most significant pathogens of nosocomial infections with high morbidity and mortality rates [5]. Despite efforts made to control this pathogen, MRSA has a high genetic adaptability and continues to produce epidemic clones that threaten public health [6].

Some *S. aureus* clonal complexes, such as CC9, CC130, and CC398, have shown the exceptional zoonotic capacity and should be closely watched because they have the potential to produce host-jumping and emerging clones in humans [7,8,9]. These include MRSA strains, whose transmission from livestock to humans has been so extensively described that it has an epidemiological category of its own: livestock-associated (LA-MRSA) [9]. Therefore, these studies indicate that livestock may serve as a reservoir of strains and genetic traits for humans. Yet when it comes to wild animals, information is more ambiguous and does not fully explain the epidemiological role played by the strains harbored by these animals. They harbor a variety of STs, including those related to humans, livestock, and multiple hosts, and some harbor resistance genes for antibiotics of clinical use [10]. Apart from serving as a reservoir, these animals can actively contribute to the development of new strains. Recently, it has been proven that *mecC-*MRSA emerged in the pre-antibiotic era as a result of the natural selection exerted by the dermatophyte *Trichophyton erinacei* when colonizing wild hedgehogs [11]. Therefore, genomic surveillance of the strains harbored by these animals would increase knowledge about their genetic variability and pathogenic potential.

In Spain, previous research has been conducted in different regions to detect and characterize *S. aureus* in wild animals [12,13]. The results from the Valencian Community (Eastern Spain) have stood out for the high *mecC*-MRSA levels found in wild rabbits [14]. This study was conducted in high-density areas of lagomorphs, where the interaction between rabbits and other animals is high, and exchange of *S. aureus* strains, especially MRSA, could take place. Hence the objectives of the present study were to: (1) examine *S. aureus* carriage in wild animals of Eastern Spain; (2) identify the STs found among different wild species; (3) compare the STs from wild animals to those from people and animals that have been in direct contact; and (4) evaluate their antibiotic resistance and virulence.

## 2. Materials and Methods

### 2.1. Sample Collection

In collaboration with the Service of Analysis, Research, and Management of Wild Animals (SAIGAS) of the Universidad Cardenal Herrera-CEU, sampling was carried out in 2021–2022 in different towns of the Valencian Community (Eastern Spain). Game species, such as red deer (*Cervus elaphus*), Eurasian wild boar (*Sus scrofa*), European mouflon (*Ovis orientalis musimon*), and Iberian ibex (*Capra pyrenaica*), were sampled during driven hunts. In addition, hunters, along with their dogs and ferrets (auxiliary hunting animals), volunteered to be sampled. The remaining species were obtained from the “La Granja del Saler” wildlife rescue center or appeared as roadkills or dead in the countryside and were collected by the SAIGAS group.

Whenever possible, an independent ear, nasal and genital swab was taken from each animal. Given nasal sampling difficulties in smaller species, a nasopharyngeal swab was used instead. For birds, no auricular swab was collected, and the nasal sample was replaced with a nasopharyngeal swab, while the genital sample was replaced with a cloacal swab. Humans were tested by a single nasal swab.

### 2.2. Staphylococcus aureus Isolation and Characterization

Samples were inoculated on colistin-nalidixic acid (CNA) blood agar (Becton-Dickinson, Sparks, MD, USA) and incubated aerobically at 37 °C for 24 h. For a second test, the colonies that met the criteria for *S. aureus* compatibility in terms of morphological growth and hemolytic characteristics in CNA were incubated in CHROMagar^TM^ Staph aureus and CHROMagar^TM^ Orientation. Final confirmation was made by the amplification of the coagulase (*coa*) [15] and protein A (*spa*) genes [16]. The Genelute Bacterial Genomic DNA kit (Sigma-Aldrich, St. Louis, MO, USA) was used to extract DNA for PCRs. In order to do so, the manufacturer’s instructions were followed, except for lysostaphin (12.5 g/mL, Sigma-Aldrich), which was used to lyse bacterial cells at 37 °C for 1 h prior to extraction. Once confirmed, isolates were genotyped according to the analysis of the polymorphic region of genes *coa* and *spa* [17].

### 2.3. Genome Sequencing and Assembly

Among the positive samples, representative isolates were selected according to their *coa*-*spa* genotype, the animal species from which they were obtained, and the number of individuals to undergo whole genome sequencing. DNA libraries were prepared using xGen DNA Library Prep EZ (IDT). Capillary electrophoresis was carried out employing the QIAxcel Advanced System (Qiagen) to assess libraries’ output. Samples were sequenced on a Hiseq XTen (Illumina) (2 × 150 bp paired-end). Some sequences were pre-processed with fastp (Version 0.23.2) [18] after quality control using FastQC (Version 0.25.1) [19] and Quast (Version 5.2.0) [20]. Sequencing data were uploaded to the Galaxy web platform to perform the subsequent genomic analysis [21]. Genomes were *de novo* assembled with SPAdes 3.15.3 using automatic k-mer selection in the assembly and error correction model [22]. The Illumina sequences generated and used in this study are deposited in the NCBI Sequence Read Archive (SRA) with BioProject accession number (PRJNA940825).

### 2.4. Molecular Typing and Phylogenetic Analysis

Multilocus sequence typing (MLST) was performed with MLST Galaxy Version 2.22.0 [23], which scans genomes against PubMLST schemes. When identified, new allele sequences were submitted to the PubMLST website to assign an allele number [24]. Similarly, when a new profile was found, it was submitted to PubMLST for an ST number. Genomes were annotated with Prokka Galaxy Version 1.14.6 [25]. The resulting output, in the gff3 format, was submitted to Roary Galaxy Version 3.13.0 to find and align core genes by default parameters [26]. From this alignment, it is possible to construct a maximum likelihood phylogenetic tree of isolates using RaxML Galaxy Version 8.2.4 20 [27]. The phylogenetic tree was visualized and annotated using the Interactive Tree of Life (iTOL) online platform [28]. Antimicrobial resistance and virulence genes were searched in the assembled genomes using ABRicate Galaxy Version 1.0.1 [29] in the Resfinder database [30], Comprehensive Antibiotic Resistance Database (CARD) [31] and the Virulence Factor Database (VFDB) [32]. A minimum of 80% nucleotide sequence identity and coverage was used to identify positive hits. *S. aureus* may harbor a wide range of virulence factors. However, only those that varied between isolates were examined. These genes could be classified into two categories: immune evasion cluster (IEC) genes, which play a role in host infection and adaptation [33], and superantigens, which induce the overproduction of cytokines and are associated with some human diseases [34]. When a *mec* gene was found, the SCC*mec* element was typed using SCC*mec*Finder [35], which is available on the Center for Genomic Epidemiology website.

### 2.5. Phenotypic Antimicrobial Susceptibility Testing

Isolates were tested for antimicrobial susceptibility by the disk diffusion method on Mueller-Hinton agar (Becton-Dickinson, Sparks, MD, USA) following the recommendations of the European Committee on Antimicrobial Susceptibility Testing breakpoints (Version 12.0) [36]. *S. aureus* ATCC 25,923 was used as a quality control strain. The antibiotics from the four categories established by the European Medicines Agency (EMA), as well as the agents used as a screening test to detect susceptibility to one antibiotic of its own category or more [36], were included in the selection criteria for antibiotics. This included frequently used antibiotics in daily practice against staphylococcal infections and those that allow the detection of resistance to antibiotics for both livestock and human medical use.

Following these criteria, the finally tested antibiotics were: benzylpenicillin (1 U), cefoxitin (30 μg), ceftaroline (5 μg), clindamycin (2 μg), chloramphenicol (30 μg), erithromycin (15 μg), gentamicin (10 μg), kanamycin (30 μg), linezolid (10 μg), norfloxacin (10 μg), tetracycline (30 μg), tigecycline (15 μg) and trimethoprim/sulfamethoxazole (1.25/23.75 μg, respectively). No disk diffusion method was available for fosfomycin and streptomycin, according to EUCAST [36]. Trimethoprim susceptibility testing was replaced with trimethoprim/sulfamethoxazole susceptibility testing because trimethoprim is frequently employed in combination with sulfonamides in clinical practice.

### 2.6. Statistical Analysis

The percentage of wild animals that tested positive for *S. aureus* was evaluated using a generalized mixed model, with a binomial probability distribution for the response and logit transformation [ln(µ/1 − µ))] as a link function (Proc Glimmix of SAS). To evaluate the effect of the zone of the isolate (3 levels; Nasal, Genital, Auricular), the model included all the isolates (*n =* 637) except for the isolates from Eurasian magpie (*Pica pica*) due to birds’ anatomical characteristics. The model included the effect of the zone as the fixed effect. In order to evaluate the species effect, only one record per animal was evaluated. An animal was considered positive if *S. aureus* was found in any of the three locations. Only the species with more than five sampled animals were used (12 species, 228 animals). The model included the species effect as the fixed effect.

The number of virulence genes, the number of resistance genes, and the number of antibiotics to which a strain was resistant were evaluated using a linear mixed model. The model included the effects of ST (30 levels), sampled species (15 levels), and error (83 levels) as random effects.

## 3. Results

### 3.1. Animal Sampling and S. aureus Identification

In all, 359 individuals were sampled, including seven hunting ferrets, 55 hunting dogs, 55 hunters, and 242 wild animals. The wild animal collection featured 20 different animal species, including the following orders: Artiodactyla, Carnivora, Chiroptera, Eulipotyphla, Passeriformes, and Rodentia (Table 1).

Of the wild species, Artiodactyla and Carnivora were by far the most sampled orders (with 83 and 82 individuals each) and had a wide variety of different sampled species (four and nine species, respectively). The rest of the orders had fewer sampled individuals and species.

*S. aureus* was detected in 28.1% (68 out of 242) of the investigated wild animals. This percentage varied among orders: 46.7% (14 out of 30) for Eulipotyphla, 42.2% (35 out of 83) for Artiodactyla, 26.7% (4 out of 15) for Chiroptera, 25% (2 out of 8) for Rodentia, 13.4% (11 out of 82) for Carnivora and 8.3% (2 out of 24) for Passeriformes (Table 1). However, the results within orders considerably varied. Figure 1 presents the percentage of positive animals per species considered for the statistical analysis. We observe that in the Artiodactyla order, there were no differences in the percentage of positive animals to *S. aureus* among species. However, in the Carnivora order, the results considerably varied. Indeed, 60% of common genets were positive, whereas only 5% of a red fox or beech marten were positive (*p* < 0.05). These results reveal that the presence of *S. aureus* depends more on species than on order. In fact, more than 50% of European hedgehogs, common genet, or Iberian ibexes were positive, whereas less than 10% of Eurasian magpies, red foxes, or beech marten were positive (*p* < 0.05). The percentage of positive animals from red squirrel, common pipistrelle, wild boar, or red deer was intermediate, with no statistical differences between these animals and those reported above.

For the hunting sector, 3.6% (2 out of 55) of dogs, 14.3% (1 out of 7) of ferrets, and 16.4% (9 out of 55) of hunters were positive for *S. aureus*.

### 3.2. Sampling Localization

As described above, whenever possible, each animal was sampled from all three locations (nasal, auricular, and genital location), and, therefore, *S. aureus* could be isolated in different locations in the same animal. *S. aureus* was found in 99 of the 911 samples that were collected from the 359 animals and humans studied in this work. Dogs, ferrets, and hunters carried *S. aureus* in the nasal localization (12 isolates) exclusively. The remaining 87 positive samples were obtained from 12 of the 20 studied wild species, as detailed in Table 1.

For the statistical analysis, which included all of the wild animals except for Eurasian magpies, *S. aureus* was located mainly in the nasal cavity (51.7%, 44 out of 85), followed by the genital (28.2%, 24 out of 85) and auricular (20.0%, 17 out of 85) locations. According to these findings, *S. aureus* is twice as likely to be isolated at nasal locations than at genital ones (*p* = 0.011) and three times as likely to be isolated in nasal locations than in auricular ones (*p* < 0.001). The nasal swab was also the main source of *S. aureus* when each species was separately considered, except for hedgehogs and genets. There were few differences in isolation in the three anatomical regions for hedgehogs (34.8%, 34.8%, and 30.4% for auricular, nasal, and genital localizations, respectively). For genets, 50% of the isolates were collected from the auricular location, while 25% were isolated from the nasal and 25% from genital locations. *S. aureus* was exclusively isolated from the genital location in the common pipistrelle and the European free-tailed bat. For Eurasian magpies, of the two positive individuals, one carried *S. aureus* in the nasopharyngeal location and the other in the cloaca.

### 3.3. MLST Results

Of the 99 *S. aureus* isolates, 73 were sequenced (61 from wild animals, nine human isolates, two dog isolates, and one ferret isolate). MLST typing evidenced 30 STs, including 10 STs not previously reported (Figure 2A). In the wild animals, the most isolated STs were ST2328 (10/61, 16.4%), ST49 (8/61; 13.1%), and ST425 (6/61, 9.8%). Except for ST2328, which was only found in Eurasian wild boar, these STs were discovered within a wide range of animal species.

The ST that was found in a wider variety of species was ST398 because it was isolated from red squirrels, beech martens, common genets, and European hedgehogs. Other STs isolated in multiple wild species were ST22 (3), ST2639 (2), ST6 (2), ST5 (2), ST7817 (2), ST49 (3), ST581 (2), and ST425 (2). A single isolate of each novel ST (ST7433, ST7812, ST7813, ST7814, ST7815, ST7816, ST7817, ST7818, ST7819, and ST7820) was detected, except ST7815, which was found in two wild boars from the same geographic area, and ST7817, which was detected in one red squirrel and one common genet from two distinct geographic locations. Both ST7815 and ST7817 were singletons, but the other novel STs were identified as a single locus variant (SLV) of ST130 (ST7812, ST7814, and ST7820), SLV of ST581 (ST7813 and ST7818), SLV of ST1 (ST7816) and SLV of ST5 (ST7819).

In all species with more than one positive animal, at least two different STs were found. The wild species that harbored the most diverse STs were the common genet (9), European hedgehog (6), and Iberian ibex (5). Co-occurrence of two strains in the same animal appeared in two European hedgehogs (nasal + genital and nasal + auricular + genital locations, respectively); two genets (nasal + auricular and nasal + genital locations, respectively); one mouflon (nasal + genital locations); one Iberian ibex (genital + auricular locations). Co-occurring strains coincided only on two occasions: in the genital and auricular locations in one European hedgehog (ST49) and in the nasal and genital locations in one mouflon (ST581). The previously cited European hedgehog that carried *S. aureus* in all three locations had the ST49 lineage on the auricular and genital swabs but ST425 on the nasal one. When STs did not match, strains were distant on the phylogenetic tree (Figure 2A): ST6 and ST2639 in the nasal and genital locations, respectively, of one genet; ST7820 and ST5826 in the nasal and auricular locations, respectively, of another genet; ST7819 and ST581 in the genital and auricular locations, respectively, of the Iberian ibex.

Regarding domestic animals, dogs harbored ST34 (CC30) and the novel ST7433 (SLV of ST1), while ferrets harbored ST425. Finally, humans carried ST5, ST26, ST45 (CC45), ST125, ST398 and ST582 (CC15).

### 3.4. Antibiotic Resistance Analysis

Two methods were used to explore antibiotic resistance: phenotypic testing by the disk diffusion method and genotypic detection via whole genome sequencing (Figure 2B).

After the phenotypic analysis of the 73 *S. aureus* isolates sequenced in this study, resistance to 8 of the 13 tested antibiotics was found. The variability observed in the number of antibiotics to which a strain was resistant was explained in 28% by the differences among STs (*p* = 0.011). This was a moderate effect for ST, which meant that some STs presented many resistance genes, whereas others presented only a few.

From the genomic analysis, 23 antibiotic-resistance genes were found, of which eight were ubiquitous in our dataset. Regardless of ST or host, nearly all the isolates carried antibiotic regulators *arlR* (73 out of 73), *mgrA* (73 out of 73), *arlS* (72 out of 73), and *mepR* (72 out of 73), and antibiotic efflux pumps *lmrS* (70 out of 73), *tet38* (70 out of 73), *norA* (71 out of 73), and *mepA* (69 out of 73). In Figure 2B, only the genes that were differential between STs are portrayed. Moreover, 46% of the variability in the total number of these differential genes was explained by differences between STs (*p* = 0.037). This was evidenced by the non-homogeneous distribution of the genes encoding antibiotic resistance throughout the various STs on the phylogenetic tree, except for all the ST130 and ST22 isolates, which harbored the *blaZ* gene, and all ST2639, CC5, ST425, and ST2328 isolates, which harbored the *fosB* gene (Figure 2B).

Regarding wild animals, susceptibility testing found 34.4% (21 out of 61) of resistant isolates to one antimicrobial or more. The group of antibiotics to which wild animal isolates were most resistant is penicillins (32.8%, 20 out of 61). A resistant phenotype was associated with the presence of *blaZ*, except for two isolates where it was not found. There were also two susceptible isolates that harbored *blaZ*. Although *norA* was present in most isolates, as mentioned above, only 8.2% of isolates exhibited norfloxacin resistance (5 out of 61).

In aminoglycosides, low resistance levels were found because only two strains belonging to ST398 and ST7820 exhibited resistance (3.3%, 2 out of 61). Yet according to our observations, only one presented either of the two detected resistance genes (ST398). For macrolides, two strains belonging to ST398 presented resistance in the disk diffusion test, and both presented one of the three genes coding for resistance to this antibiotic (3.3%, 2 out of 61). However, *erm*C was present in 18 more strains that were sensitive to macrolides by susceptibility testing. Similarly, the amphenicol resistance gene was present in only one ST2767 strain but without a correlative phenotype. Regarding tetracyclines, three strains belonging to ST22, ST398, and ST2767 reported a resistant phenotype (4.9%, 3 out of 61), but only one (ST398) harbored *tetK*, *tetM,* and *tetU*. Eighteen susceptible strains also carried *tetK* (18%, 11 out of 61), *tetU* (8.2%, 5 out of 61), or both (3.3%, 2 out of 61).

According to both genomic detection and susceptibility testing, five isolates recovered from wildlife were classified as MRSA, four of which carried *mecA* and only one carried *mecC*. The *mecA*-MRSA isolates were recovered from beech martens, common pipistrelles, European free-tailed bats, and European hedgehogs. Except for the ST398 isolate, which was type V, they were all classified as SCC*mec* type IV. Interestingly, the *mecA* gene was present in all the isolated strains belonging to ST22. The only *mecC*-MRSA strain (SCC*mec* type XI) was found in a common genet and belonged to ST49.

The genes that conferred resistance to diaminopyrimidines (*dfrk*) and streptomycin (*str*) were infrequent in wildlife: 1.6% (1 out of 61) and 6.6% (4 out of 61) of the strains, respectively. However, the fosfomycin resistance gene (*fosB*) was present in 42.6% (26 out of 61). No resistance to tigecycline, linezolid, trimethoprim/sulfamethoxazole, and ceftaroline was detected.

When the antibiotic groups are evaluated together, a strain is considered to be multidrug-resistant if it is resistant to three antibiotic groups or more. Three isolates (4.9%, 3 out of 61) exhibited a multidrug-resistance pattern: one MSSA-ST398 isolate recovered from a European hedgehog, one *mecA*-MRSA-ST398 recovered from a beech marteen, and one *mecA*-MRSA-ST22 recovered from a European hedgehog.

All the human strains showed resistance at both the genomic and phenotypic levels. Similarly, to wild animals, penicillin was the group of antibiotics to which the most resistance was found according to susceptibility testing (66.7%, 6 out of 9). Except for the ST125 isolate, all the penicillin-resistant isolates harbored *blaZ*. In contrast, one susceptible ST398 isolate also harbored *blaZ*. Macrolide resistance was also frequent (44.4%, 4 out of 9), and all the resistant isolates had different combinations of the following resistance genes: *ermC*, *ermT,* and *msrA*. There was also one ST5 susceptible isolation that carried *ermC*. Norfloxacin resistance was found in two isolates belonging to ST5 and ST125 (22.2%, 2 out of 9), but they all carried *norA*. Only one resistant isolate (ST125) was detected for aminoglycosides (11.1%, 1 out of 9), which carried *ANT(4’)-Ib*. Tetracycline resistance was not found, although two isolates (ST5 and ST398) had *tetK,* and three carried *tetU* (ST45, ST125, and ST398). No genomic or phenotypic resistance was detected for amphenicols. The frequency of the *str* and *fosB* genes in the human isolates was 22.2% (2 out of 9) and 55.5% (5 out of 9), respectively. Only one *mecA*-MRSA strain (ST125) was detected, which was typed as SCC*mec* type IV. It also exhibited resistance to aminoglycosides, macrolides, and norfloxacin (multidrug resistance pattern).

The two isolates obtained from dogs were considerably different: the ST7433 isolate lacked any genomic or phenotypic resistance, except for ubiquitous genes, whereas ST34 showed resistance to aminoglycosides and penicillins in susceptibility testing, in addition to carrying *blaZ* and *fosB*. The ferret isolate (ST425) did not show any phenotypic resistance but carried genes *ermC* and *fosB*.

### 3.5. Virulence Genes Analysis

We found 12 virulence factors that were differential between strains (Figure 2C). They were classified into two categories: immune evasion cluster (IEC) genes [33] and superantigens [34]. According to our findings, differences between STs accounted for 64.5% of the variability in the number of virulence genes (*p* = 0.003). This was a strong effect of ST and meant that some STs presented many virulence genes, whereas others presented a few. This was evidenced by their distribution in divergent phylogeny parts (Figure 2C).

The IEC, which is constituted by *scn*, *chp*, *sak,* and *sea* (or *sep*), can be divided into variants, each of which includes *scn* and different combinations of the other genes. Ten wild animal isolates (14.8%, 9 out of 61) carried IEC, including common pipistrelle (ST22), European free-tailed bat (ST22), European hedgehog (ST22, ST398, and ST425), Iberian ibex (ST2639 and ST7819), red fox (ST7816) and red deer (ST15). The three *mecA*-MRSA-ST22 isolates had IEC-type B, while the remaining isolates were typed as C, D, H, or as a novel type. The novel variety, carrying genes *scn*, *chp*, and *sea*, was detected in an ST15 isolate recovered from a red deer. The IEC was present in all human isolates, which were typed as A, B, or C. However, the ferret did not carry IEC genes, and only one of the two positive dogs did (ST34) and was typed as B.

Superantigen genes were found in 26.2% (16 out of 61) of the isolates, including exfoliative toxin B (*etb*), toxic shock syndrome toxin (*tsst-1*), and enterotoxins (*sea*, *sec*, *sed*, *seh*, *selk*, sell, and *selq*). In our dataset, *etb* and *tsst-1* were infrequent: only two ST130 isolates carried *etb,* and the two isolates belonging to ST2328 and ST7816 harbored the *tsst-1* gene (3.3%, 2 out of 61 for each gene). Enterotoxins were found in 13 isolates, which harbored between 1 and 2 enterotoxins per isolate, except for the ST7816 isolate, which presented four of them. When considered individually, the *sea* was the most frequent enterotoxin (11.5%, 7 out of 61), followed by *sec* (8.2%, 5 out of 61), *seh,* and *sell* (4.9%, 3 out of 61 for each) and *sed* (3.3%, 2 out of 61). Only one isolated harbored *selk* and *selq* (1.6%, 1 out of 61) along with *sea* and *seh* (ST7816). This isolate from a red fox was notable for its superantigen content and also for the presence of both enterotoxins and *tsst-1*. Neither *etb* nor *tsst-1* were detected in the human isolates, although between 1 and 2 enterotoxins per isolate were discovered in 44.4% (4 out of 9) of the strains. The most frequent enterotoxin was the *sea* (33.3%, 3 out of 9), while *sec*, *sed*, *seh*, and *sell* were detected only in one isolate for each one (11.1%, 1 out of 9). No superantigens were found in ferrets. Both dog isolates (ST34 and ST7433) had enterotoxin *seh,* although the ST34 isolate also carried *tsst-1*.

## 4. Discussion

In the present work, the *S. aureus* isolates from wild animals of Eastern Spain are described and characterized. The global carriage rate was high (27.6%) but vastly differed when each species was taken into account separately. Abdullahi et al. [37] examined 33 studies about *S. aureus* in the nasal, tracheal, or oral cavities of wild animals in Africa, America, Asia, and Europe. They found that the pooled prevalence was 18.5%. There are discrepancies when the carriage rates of each specific species are compared across studies. These differences may be attributed to the fact that *S. aureus* circulation dynamics are specific to each country or region and also to methodological differences among studies.

Although *S. aureus* was not reported in some wild species, others stood out for their high isolation, such as common genets, Iberian ibexes, European hedgehogs, red deer, and Eurasian wild boar. The common genet was particularly interesting given the high *S. aureus* carriage, the detection of one *mecC*-MRSA strain, and the wide range of STs that it harbored. To the authors’ knowledge, *S. aureus* has never been described as this species and may require further research. However, statements about the carrier rate obtained in this work should be cautiously taken given the study limitations in sampling method terms. Wildlife peculiarity includes its difficult access and, therefore, it is frequently restricted to “opportunistic sampling” because samples originate from either report of passive animal deaths or the sampling conducted during driving hunts [38].

In humans and ruminants, nares constituted the primary *S. aureus* reservoir [2] and based on our findings, this sample can be suggested for *S. aureus* screening in wildlife. However, it seems appropriate to test more than one location because many animals would not have been regarded as positive if only a nasal swab had been taken [12,13,39]. It has also been noted that multiple sampling increases the genetic diversity of *S. aureus* when identified in an animal [39].

The genomic structure of the population of the sequenced isolates showed a significant degree of diversity. Some of the lineages discovered in this study were associated with domestic animals or humans, while others were completely new and could be related to wildlife. The ST2328 lineage has been mainly identified in wild animals from the Iberian Peninsula, especially wild boars [39,40]. Clonal complex 49 (CC49), which includes ST49, is isolated mostly from wild animals and usually exhibits a methicillin-sensitive phenotype [41,42], but occasionally, as in this study, *mecC*-MRSA has been found in animals [43,44] and humans [45]. ST581 appears to be frequent in Iberian ibex [39], but a single isolate from goat’s milk is described in the MLST database [46]. Other STs that could be related to wildlife were ST2639, ST2767, ST3237, and ST5826 because both the present study and the literature describe them in a variety of wild species [14,39,41].

Three of the STs commonly described in the animal-human interface were found: ST130, ST398, and ST425. These lineages are especially interesting because of their non-specificity and host jumping ability. ST130 (CC130) and ST425 (CC425) have been identified from a wide range of animal species and are frequently observed STs in wildlife research works [10]. Both are closely related to the *mecC* gene, as most *mecC*-MRSA strains belong to these clonal complexes [47]. However, all the strains belonging to these sequence types in the present study were MSSA. The ability of CC130 to transition across several host species has been demonstrated by the numerous spillover events from humans and sheep that lead to the contemporary bovine isolates of CC130 [48]. In fact, Loncaric et al. [49] found the same strain in a wild hare, a cow, and a sheep sharing the same habitat, which suggests an exchange may occur.

There are two distinct clades in CC398, and both were represented in our samples: the human clade characterized by MSSA strains carrying IEC genes, and the animal clade characterized by strains that are IEC-negative, carry *tetM* and are MRSA in most cases [9]. As MSSA-ST398 strains have been identified as a growing clade of invasive infections in Spanish hospitals, the occurrence of these strains in wild animals may be of epidemiological interest [50]. Similar strains were found in Spanish wildlife by Gómez et al. [51], who reported that white storks exposed to human residues were frequent carriers of MSSA-CC398.

We also found some human-associated STs in wild animals: ST1, ST5, and ST22 [52]. All the ST22 isolates were *mecA*-MRSA and were isolated in two bat species and one European hedgehog. ST22 is considered one of the major HA-MRSA clones, and the presence of IEC genes in our isolates indicates that these animal strains may be of human origin [53]. ST5 is another lineage of human origin that has successfully extended to livestock species, such as poultry and swine [54,55]. New mobile genetic components were acquired as a result of adaptation to these hosts, while the IEC and other genes implicated in human pathogenesis were lost. IEC is carried by φSa3int phages, and its mobilization plays a key factor in *S. aureus* host infection, switching, and adaptation in human and animal hosts [33]. This might explain why the ST5 samples from humans carried the IEC genes but not those from magpies or wild boars. Along with ST5, ST6 is a member of CC5 and a human lineage that has been isolated in domestic animals and nonhuman primates [56]. Finally, we collected isolates from CC1, a traditionally human lineage that has also been reported in wild animals [41].

The diversity of STs circulating in wildlife raises the question of these strains’ origin and evolution. Some lineages carried the human genetic marker IEC and, therefore, were most likely to be of human origin. These strains may be transmitted to wild animals by human sources, such as sewage effluents, which may end up in rivers. Frequently, animal and human wastewater are discharged into rivers [57], and, despite being treated before being released, research has shown that *S. aureus* still persists after treatment [58,59]. As a result, surface watercourses may be contaminated, which might lead to the spread of *S. aureus* to nearby wildlife and the environment. In Spain, the same *mecC*-MRSA strain has been found in a river and in wild animals from the same area, which poses the possibility that water may serve as an MRSA reservoir [60]. Therefore, it would seem appropriate to include multiple environmental samplings in the epidemiological monitoring of *S. aureus* in the future to better identify the sources that are critical for the bacterium to spread.

Transmission can also occur when domestic animals or people come into contact with wildlife and the other way around. For example, red deer and Eurasian wild boar are primary big game species, and certain regions maintain high-density populations of these species [61]. There is evidence that these populations share resources with extensive farms [62], which could facilitate the exchange of *S. aureus* strains in the environment. The increase in wild boar populations and this species’ good adaptability have led to wild boars’ opportunistic foraging and feeding behavior near human settings, which encourages contact with human populations [63]. However, most of the isolates recovered from these species belonged to wildlife-associated lineages and lacked IEC genes.

Auxiliary game animals, such as dogs and ferrets, are frequently used by hunters to improve hunting efficiency, and they come in close proximity to different wild species. Ferrets are used to access rabbit burrows. In our study, only one ferret tested positive for *S. aureus*. It harbored the ST425 lineage, which is, to the authors’ knowledge, the first report of an *S. aureus*’ ST in ferrets. ST425 is very common in wild rabbits in the Valencian community [14], which suggests an exchange of strains. Yet as information on ferret genetic lineages is lacking, the absence of sequenced rabbit strains in our study and the multispecies nature of ST425, this hypothesis should be confirmed by further studies. Conversely, dogs did not share STs with either hunters or wild animals, but both STs belonged to the clonal complexes associated with humans (CC1 and CC30). This seems consistent with the main *S. aureus* colonization hypothesis in dogs insofar as humans act as the primary source of *S. aureus* strains for dogs [1]. One of the isolates also carried the human marker IEC, which is a typical finding in MSSA isolates from dogs and could result from community or dog owner transmission [64].

All the STs found in hunters belong to lineages that are commonly isolated from humans. However, the strains belonging to ST5 and ST398 were found in both hunters and animals. Some *S. aureus* is able to colonize different hosts, which means that their genes can spread more widely because they are not restricted to the animal host that harbors them [65]. In some cases, phylogenetic proximity between isolates can facilitate the exchange of genetic material [66,67]. Therefore, the isolates obtained from wild animals, particularly those belonging to human lineages, may serve as a genetic reservoir. We found that the superantigen and antibiotic resistance genes were distributed among isolates from different animal hosts and STs. This resulted in the distribution of these genes on many phylogenetic branches, which suggests that numerous horizontal gene transfer events could have occurred [65]. Future research should examine whether these genes are indeed located in mobile genetic elements. Moreover, further investigation of the mobile genetic elements of these strains would be useful for determining the degree of adaptation to these hosts as well as for comprehending the epidemiological role they play in *S. aureus* transmission.

Superantigens are a kind of virulence factors that are particularly important because some of them cause significant diseases in humans: *etb* is one of the exfoliative toxins involved in staphylococcal scalded skin syndrome (SSSS) [34]; food contamination by enterotoxins is one of the main causes of food poisoning outbreaks [68]; and *tsst-1* is responsible for the toxic shock syndrome, a severe systemic disease that can lead to multi-organ failure [69]. We also studied the IEC, which was present in different STs and animals, and suggested phage horizontal transmission between strains. If transmission of the phages involved in host tropism occurs in nature, then the genomic surveillance of these animal strains is even more necessary because even specialized STs would have the potential to acquire or lose IEC genes and to emerge as an endemic lineage in cattle and humans [33].

Studies on antimicrobial resistance in wildlife report different prevalences depending on the animal species and the studied geographical region. According to our data, 34.4% of the isolates showed resistance to at least one antimicrobial agent. This is a rather high percentage considering that these animals were not apparently in direct contact with antibiotics. In particular, resistance to penicillin stood out in 32.8% of the isolates. This seems to be the most frequent resistance in wildlife [37]. A similar research work, carried out in Spain with several wild species, revealed an overall MRSA frequency of 0.89% [39]. In our study, methicillin resistance was 8.2% of strains, which was lower than expected based on the results of Moreno-Grúa et al. [14] obtained in the same geographical area (63.3%). In addition, in that study, almost all the strains were *mecC-*MRSA and belonged to CC130, whereas only one *mecC*-MRSA-ST49 strain was found in our study. The recurrent detection of *mecC*-MRSA lineages in wild animals may indicate that it is associated with wildlife [47]. In fact, the origin of the majority of the *mecC*-MRSA lineages has been recently found to lie in hedgehogs. Nevertheless, it is possible that certain domestic animals might serve as intermediate hosts in zoonotic transmission to humans [11,70]. However, all those tested in this study were MSSA, except for one hedgehog that carried the *mecA* gene.

When evaluating the resistance concordances between genotype and phenotype, whole genome sequence analysis showed certain limitations. Discrepancies might result from biological causes such as modifications in regulatory regions, alternative resistance mechanisms, or different gene expression patterns, or they could result from possible sequencing errors [71]. This is why many authors agree that susceptibility testing will remain the standard procedure until nucleotide and protein databases’ algorithms are able to identify all the variations and crucial elements linked with resistance patterns [72,73].

## 5. Conclusions

In summary, this study has focused on the genetic characterization of *S. aureus* in wildlife. Our findings highlight the role of wildlife as a reservoir for some clinically relevant *S. aureus* strains and genes, including isolates with IEC genes, resistance genes (*blaZ* and SCC*mec*, among others) for the antibiotics used in clinical practice, as well as superantigens (enterotoxins, *etb,* and *tsst-1*) implicated in the pathogenicity of certain human *S. aureus* infections. Interactions between the environment, domestic and wild animals and humans influence the transmission dynamics and evolution of *S. aureus* lineages and genes. Since we have found certain genes and lineages in wildlife that have traditionally been associated with humans, additional measures need to be taken to identify and prevent transmission pathways between humans and wildlife.

## Figures and Tables

**Figure 1 animals-13-01064-f001:**
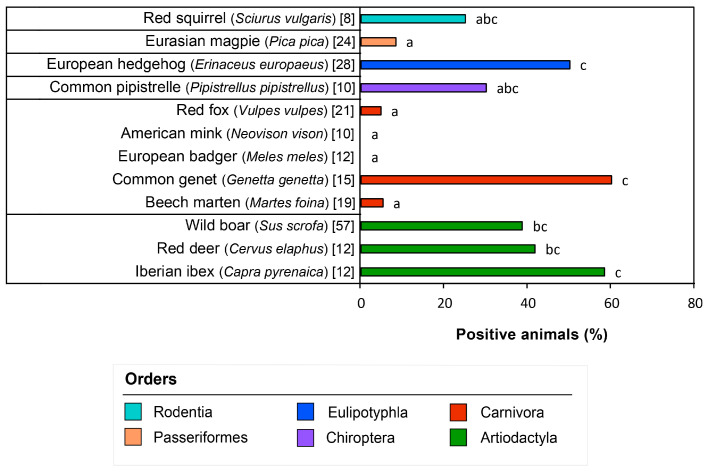
The percentage is of animals positive to *S. aureus* for the different examined wild species. We only considered species for which we sampled more than five animals. The bars not sharing the same letter differ significantly (*p* < 0.05).

**Figure 2 animals-13-01064-f002:**
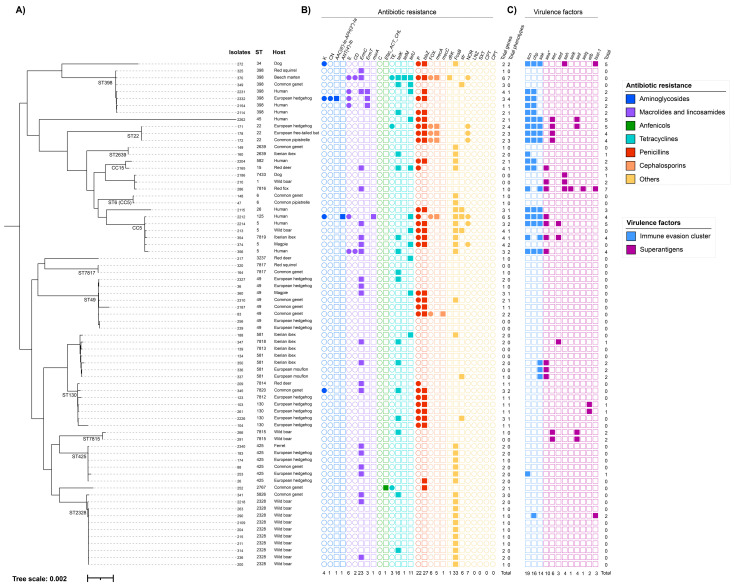
Phylogenetic diversity, antibiotic resistance, and virulence of the 73 *S. aureus* isolate sequenced in this study. (**A**) Midpoint-rooted maximum likelihood phylogenetic tree constructed using core genes within genome sequences. The tree scale represents the number of substitutions per site (0.002). If more than one isolate belonged to an ST, branches were labeled by the ST. Otherwise, they were labeled by the CC when numerous STs belonged to the same clonal group. Isolates’ names, animal species, and STs are shown in the tips. (**B**) Phylogenetic distribution of genomic and phenotypic resistance. The presence/absence profile of each isolate for 15 antibiotic resistance genes are represented by squares (filled square, present; outlined square, absent). The resistant/susceptible phenotype of antibiotic susceptibility testing for 13 antibiotics is represented by circles (filled circle, resistant; outlined circle susceptible). The legend for each gene and antibiotic appears at the top of each vertical line. The color of each symbol depends on the family of antibiotics: blue for aminoglycosides; dark purple for macrolides and lincosamides; light purple for amphenicols; pink for tetracyclines; red for penicillins; orange for cephalosporins; and yellow for the genes and antibiotics from different families of antimicrobials. An indication of each isolate’s total number of resistance genes and antibiotics to which it is resistant is included at the end of each horizontal line. The total number of isolates that harbor a particular gene or resistance is indicated at the end of each vertical line. K, kanamycin; CN, gentamycin; E, erythromycin; CD, clindamycin; C, chloramphenicol; TE, tetracycline; P, benzylpenicillin; FOX, cefoxitin; NOR, norfloxacin; TGC, tigecycline; LNZ, linezolid; SXT, trimethoprim-sulfamethoxazole; CPT, ceftaroline. (**C**) Phylogenetic distribution of virulent factors. The presence/absence profile of each isolate for 12 virulence genes are represented by squares (filled square, present; outlined square, absent). The legend for each gene is at the top of each vertical line. The color of each symbol depends on the virulence gene category: blue for the immune evasion cluster (IEC) and purple for superantigens. An indication of each isolate’s total number of virulence genes is included at the end of each horizontal line. The total number of isolates harboring a particular gene is indicated at the end of each vertical line. * *sea* is both a superantigen and part of the IEC.

**Table 1 animals-13-01064-t001:** Positive wild animals were obtained in this study.

Order/Species.	Animals (Total)	Nº and % of Positive Animals to *S. aureus*	Auricular Isolates	Nasal Isolates	Genital Isolates	Total Isolates
**Artiodactyla**	**83**	**35 (42.2%)**				
Eurasian wild boar (*Sus scrofa*)	57	22 (38.6%)	0	20	4	24
European mouflon (*Ovis orientalis musimon)*	2	1 (50%)	0	1	1	2
Iberian ibex (*Capra pyrenaica*)	12	7 (58.3%)	2	4	3	9
Red deer (*Cervus elaphus*)	12	5 (41.7%)	0	4	1	5
**Carnivora**	**82**	**11 (13.4%)**				
American mink (*Neovison vison*)	10	0 (0%)	-	-	-	-
Beech marten (*Martes foina*)	19	1 (5.3%)	0	1	0	1
Common genet (*Genetta genetta*)	15	9 (60%)	6	3	3	12
Eurasian otter (*Lutra lutra*)	3	0 (0%)	-	-	-	-
European badger (*Meles meles*)	12	0 (0%)	-	-	-	-
European polecat (*Mustela putorius*)	1	0 (0%)	-	-	-	-
European wildcat (*Felis silvestris*)	1	0 (0%)	-	-	-	-
Red fox (*Vulpes vulpes*)	21	1 (4.8%)	0	1	0	1
**Chiroptera**	**15**	**4 (26.7%)**				
Common pipistrelle (*Pipistrellus pipistrellus*)	10	3 (30%)	0	0	3	3
European free-tailed bat (*Tadarida teniotis)*	3	1 (33.3%)	0	0	1	1
Soprano pipistrelle (*Pipistrellus pygmaeus)*	2	0 (0%)	-	-	-	-
**Eulipotyphla**	**30**	**14 (46.7%)**				
European hedgehog (*Erinaceus europaeus*)	28	14 (50%)	8	8	7	23
Greater white-toothed shrew (*Crocidura russula*)	1	0 (0%)	-	-	-	-
North African hedgehog (*Atelerix algirus*)	1	0 (0%)	-	-	-	-
**Passeriformes**	**24**	**2 (8.3%)**				
Eurasian magpie (*Pica pica*)	24	2 (8.3%)	-	1 *	1 *	2
**Rodentia**	**8**	**2 (25%)**				
Red squirrel *(Sciurus vulgaris)*	8	2 (25%)	1	2	1	4
**Total**	**242**	**68 (28.1%)**	**17**	**45**	**25**	**87**

(*) Due to birds’ anatomical characteristics (*Pica pica*), the samples from the nasal and genital locations were substituted for samples obtained from oropharyngeal and cloacal locations, respectively. The names in bold correspond to the name of the order that includes the species indicated in the following rows in light characters. The total of each column is also shown in bold in the last row.

## Data Availability

Not applicable.

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
