# Peer review of "Genomic Characterization of Staphylococcus aureus in Wildlife"

_animals, 2023, doi:10.3390/ani13061064_

Round 1
Reviewer 1 Report
The article sent by Martínez-Seijas et al. describes the isolation and characterization of Staphylococcus aureus in wildlife. S aureus is a bacterium of great importance due to its sanitary and socio-economic consequences and therefore the study of its epidemiology is crucial. Wildlife has been described as a key component in the circulation of pathogens between the environment and domestic animals. Therefore, this study focused from a One Health approach is necessary to understand the dynamics of the different S. aureus serotypes.
In general, the writing of the article is very good, with an almost perfect level of English. However, I would like to make some suggestions to the authors to improve the quality:
- L16: "The ability of livestock to serve as a reservoir for human strains and genetic traits is widely acknowledged". I suggest that the authors reformulate this sentence since, although it is true that livestock can act as a reservoir, it cannot be considered a skill as such since it is not trained.
- L135: just a note to remind authors to add the accession code.
- L192: seems to be missing a space after the point.
- L226-233: line 230 indicates "The remaining 87 were obtained from 12 of the 20 studied wild species as detailed in Table 1". However, Table 1 shows a total of 68 samples obtained from wild fauna... Are they 87 or 68? On the other hand, line 233 indicates another different total n that does not coincide with any of the previous ones (51.7%, 44 out of 85). Which one is correct?
- L281: "was ST398" is repeated.
- L323-325: this paragraph is repeated, please delete it.
- L394-399: in my opinion, this information should be included in the material and methods, not in the results.
- L475: there is a double space after reference 44.
On the other hand, this study presents a large number of results, which demonstrates the hard work performed by the researchers and its scientific quality. I really liked Figure 2 as it describes each of the isolates at the genetic level. However, I think that due to the number of results described in the text, it is easy to get lost, and reading and understanding this section becomes hard. So, I would suggest that the authors include part of the results in a table or figure to make the text easier to read.
Reviewer 2 Report
The study is scientifically relevant and technically sound. In order to improve the text for publication, my suggestions are the following:
Lines 53-55: Please replace the cited reference (1; it is a review) for the original study from where the information was published. “It has been reported that less specialized strains may act as a reservoir of new virulence factors and antibiotic resistances because they are less adapted and it is easier to jump between hosts [1].”
Line 67: please state that CC are clonal complexes (set of clones), not single clones.
Line 72-73: the sentence lacks one or more references.
Line 80: add a space between “hedgehogs[11]”
Item 2.2 (Staphylococcus aureus isolation and characterization; line 113): how the authors are sure that the isolates are not other coagulase-positive Staphylococci (e.g. Staphylococcus intermedius group), commonly found in animals? Those other species may grow like S. aureus in the CHOMO agar and harbor the coa gene.
Item2.3 (Genome sequencing and assembly; line 124): the authors should state how they evaluated the quality of the sequencing and of the assembly. How can they guarantee the genomes are good enough for the further analysis? This is a concern since even minor sequencing errors could lead to an incorrect classification of the ST, since only one incorrect base can change the classification.
Lines 147-151: please include the version used of the databases.
Improve quality/resolution of the figure 2.
Lines 323-325: the sentence is repeated on lines 307-309
Line 475: remove the extra space between “[44] found”
Lines 544-546: Complementarily, I suggest adding that it is important to assess whether resistance and virulence genes are located in MGE, as this fact would increase the risk of dissemination among species.
Lines 574-580: Sequencing errors could also limit the WGS analysis, specially MLST but also the distribution of the resistance/virulence genes.
Lines 581-597 (conclusion): The conclusion is written as if it were just a summary of the results. Consider re-writing this topic with real conclusions (consequences, contributions, perspectives).
